# Reinforcement learning via replica stacking of quantum measurements for the training of quantum Boltzmann machines

## Abstract

Recent theoretical and experimental results suggest the possibility of using current and near-future quantum hardware in challenging sampling tasks. In this paper, we introduce free-energy-based reinforcement learning (FERL) as an application of quantum hardware. We propose a method for processing a quantum annealer's measured qubit spin configurations in approximating the free energy of a quantum Boltzmann machine (QBM). We then apply this method to perform reinforcement learning on the grid-world problem using the D-Wave 2000Q quantum annealer. The experimental results show that our technique is a promising method for harnessing the power of quantum sampling in reinforcement learning tasks.

## 1 Introduction

Reinforcement learning (RL) Sutton & Barto (1998); Bertsekas & Tsitsiklis (1996) has been successfully applied in fields such as engineering Derhami et al. (2013); Syafiie et al. (2007), sociology Erev & Roth (1998); Shteingart & Loewenstein (2014), and economics Matsui et al. (2011); Sui et al. (2010). The training samples in reinforcement learning are provided by the interaction of an agent with an ambient environment. For example, in a motion planning problem in uncharted territory, it is desirable for the agent to learn to correctly navigate in the fastest way possible, making the fewest blind decisions. That is, neither exploration nor exploitation can be pursued exclusively without either facing a penalty or failing at the task. Our goal is, therefore, not only to design an algorithm that eventually converges to an optimal policy, but for the algorithm to be able to generate suboptimal policies early in the learning process.

Free-energy-based reinforcement learning (FERL) using a restricted Boltzmann machine (RBM), as suggested by Sallans & Hinton (2004), relies on approximating a utility function for the agent, called the *Q-function*, using the free energy of an RBM. RBMs have the advantage that their free energy can be efficiently calculated using closed formulae. RBMs can represent any joint distribution over binary variables Martens et al. (2013); Hornik et al. (1989); Le Roux & Bengio (2008); however, this property of universality may require exponentially large RBMs Martens et al. (2013); Le Roux & Bengio (2008).

General Boltzmann machines (GBM) are proposed in an effort to devise universal *Q*-function approximators with polynomially large Boltzmann networks Crawford et al. (2018). Traditionally, Monte Carlo simulation is used to perform the computationally expensive tasks of approximating the free energy of GBMs under a Boltzmann distribution. One way to speed up the approximation process is to represent a GBM by an equivalent physical system and try to find its Boltzmann distribution. An example of such a physical system is a quantum annealer consisting of a network of pair-wise interacting quantum bits (qubits). Although quantum annealers have already been used in many areas of computational science, including combinatorial optimization and machine learning, their application in RL has not been explored.

In order to use quantum annealing for RL, we first represent the *Q*-function as the free energy of a physical system, that is, that of a quantum annealer. We then slowly evolve the state of the physical system from a well-known initial state toward a state with a Boltzmann-like probability distribution. Repeating the annealing process sufficiently long can provide us with samples from the Boltzmann distribution so that we can empirically approximate the free energy of the physical system under this

distribution. Finally, approximating the free energy of the system would give us an estimate of the $Q$-function.

Up until the past few years, studies were limited to the classical Boltzmann machines.[1] Recently, Crawford et al. (2018) generalized the classical method toward a quantum or quantum-inspired algorithm for approximating the free energy of GBMs. Using simulated quantum annealing (SQA) Crawford et al. (2018) showed that FERL using a deep Boltzmann machine (DBM) can provide a drastic improvement in the early stages of learning, yet performing the same procedure on an actual quantum device remained a difficult task. This is because sampling from a quantum system representing a quantum Boltzmann machine is harder than the classical case, since at the end of each anneal the quantum system is in a superposition. Any attempt to measure the final state of the quantum system is doomed to fail since the superposition would collapse into a classical state that does not carry the entirety of information about the final state.

In this work, we have two main contributions. We first employ a quantum annealer as a physical device to approximate the free energy of a classical Boltzmann machine. Second, we generalize the notion of classical Boltzmann machines to quantum Boltzmann machines within the field of RL and utilize a quantum annealer to approximate the free energy of a quantum system. In order to deal with the issue of superposition mentioned above, we propose a novel stacking procedure in that we attempt to reconstruct the full state of superposition from the partial information that we get from sampling after each anneal. Finally we report proof-of-concept results using the D-Wave 2000Q quantum processor to provide experimental evidence for the applicability of a quantum annealer in reinforcement learning as predicted by Crawford et al. (2018).

## 2 PRELIMINARIES

We refer the reader to Sutton & Barto (1998) and Yuksel (2016) for an exposition on Markov decision processes (MDP), controlled Markov chains, and the various broad aspects of reinforcement learning. A *Q-function* is defined by mapping a tuple $(\pi, s, a)$ of a given *stationary policy* $\pi$, a current state $s$, and an immediate action $a$ of a controlled Markov chain to the expected value of the instantaneous and future discounted rewards of the Markov chain that begins with taking action $a$ at initial state $s$ and continuing according to $\pi$:

$$Q(\pi, s, a) = \mathbb{E}[r\,(s,\,a)] + \mathbb{E}\left[\sum_{i=1}^{\infty} \gamma^i\, r\left(\Pi_i^s,\, \pi(\Pi_i^s)\right)\right].$$

Here, $r(s, a)$ is a random variable, perceived by the agent from the environment, representing the immediate reward of taking action $a$ from state $s$, and $\Pi$ is the Markov chain resulting from restricting the controlled Markov chain to the policy $\pi$. The fixed real number $\gamma \in (0, 1)$ is the *discount factor* of the MDP. From $Q^*(s, a) = \max_\pi Q(\pi, s, a)$, the optimal policy for the MDP can be retrieved via

$$\pi^*(s) = \mathrm{argmax}_a\, Q^*(s, a). \tag{1}$$

This reduces the MDP task to that of computing $Q^*(s, a)$. Through the Bellman optimality equation Bellman (1956), we get

$$Q^*(s, a) = \mathbb{E}[r\,(s,\,a)] + \gamma \sum_{s'} \mathbb{P}(s'|s, a) \max_{a'} Q^*(s', a'), \tag{2}$$

so $Q^*$ is the fixed point of the following operator defined on $L_\infty(S \times A)$:

$$T(Q) : (s, a) \mapsto \mathbb{E}[r\,(s,\,a)] + \gamma \int \max_{a'} Q\,.$$

In this paper, we focus on the TD(0) $Q$-learning method, with the $Q$-function parametrized by neural networks in order to find $\pi^*(s)$ and $Q^*(s, a)$, which is based on minimizing the distance between $T(Q)$ and $Q$.

---

[1]In this paper, restricted, deep, and general Boltzmann machines are referred to as *classical* Boltzmann machines to indicate the contrast with *quantum* Boltzmann machines.

## 2.1 CLAMPED BOLTZMANN MACHINES

A clamped Boltzmann machine is a GBM in which all visible nodes $\mathbf{v}$ are prescribed fixed assignments and removed from the underlying graph. Therefore, the energy of the clamped Boltzmann machine may be written as

$$\mathcal{H}_{\mathbf{v}}(\mathbf{h}) = -\sum_{v \in V, h \in H} w^{vh} vh - \sum_{\{h,h'\} \subseteq H} w^{hh'} hh' , \tag{3}$$

where $V$ and $H$ are the sets of visible and hidden nodes, respectively, and by a slight abuse of notation, the letter $v$ stands both for a graph node $v \in V$ and for the assignment $v \in \{0, 1\}$. The interactions between the variables represented by their respective nodes are specified by real-valued weighted edges of the underlying undirected graph represented by $w^{vh}$, and $w^{hh'}$ denotes the weights between visible and hidden, or hidden and hidden, nodes of the Boltzmann machine, respectively.

A clamped quantum Boltzmann machine (QBM) has the same underlying graph as a clamped GBM, but instead of a binary random variable, qubits are associated to each node of the network. The energy function is substituted by the quantum Hamiltonian of an induced transverse field Ising model (TFIM), which is mathematically a Hermitian matrix

$$\mathcal{H}_{\mathbf{v}} = -\sum_{v \in V, h \in H} w^{vh} v \sigma_h^z - \sum_{\{h,h'\} \subseteq H} w^{hh'} \sigma_h^z \sigma_{h'}^z - \Gamma \sum_{h \in H} \sigma_h^x , \tag{4}$$

where $\sigma_h^z$ represent the Pauli $z$-matrices and $\sigma_h^x$ represent the Pauli $x$-matrices. Thus, a clamped QBM with $\Gamma = 0$ is equivalent to a clamped classical Boltzmann machine. This is because, in this case, $\mathcal{H}_{\mathbf{v}}$ is a diagonal matrix in the $\sigma^z$-basis, the spectrum of which is identical to the range of the classical Hamiltonian (3). We note that (4) is a particular instance of a TFIM.

## 2.2 FREE-ENERGY-BASED REINFORCEMENT LEARNING

Let us begin with the classical Boltzmann machine case. Following Sallans & Hinton (2004), for an assignment of visible variables $\mathbf{v}$, $F(\mathbf{v})$ denotes the *equilibrium free energy*, and is given via

$$F(\mathbf{v}) = \sum_{\mathbf{h}} \mathbb{P}(\mathbf{h}|\mathbf{v}) \mathscr{E}_{\mathbf{v}}(\mathbf{h}) + \frac{1}{\beta} \sum_{\mathbf{h}} \mathbb{P}(\mathbf{h}|\mathbf{v}) \log \mathbb{P}(\mathbf{h}|\mathbf{v}) \tag{5}$$

$$= -\sum_{\substack{s \in S \\ h \in H}} w^{sh} s \langle h \rangle - \sum_{\substack{a \in A \\ h \in H}} w^{ah} a \langle h \rangle - \sum_{\{h,h'\} \subseteq H} u^{hh'} \langle hh' \rangle$$

$$+ \frac{1}{\beta} \sum_{\mathbf{h}} \mathbb{P}(\mathbf{h}|\mathbf{s}, \mathbf{a}) \log \mathbb{P}(\mathbf{h}|\mathbf{s}, \mathbf{a}),$$

where $\beta = \frac{1}{k_B T}$ is a fixed thermodynamic beta. In Sallans & Hinton (2004), it was proposed to use the negative free energy of a GBM to approximate the $Q$-function through the relationship

$$Q(s, a) \approx -F(\mathbf{s}, \mathbf{a}) = -F(\mathbf{s}, \mathbf{a}; \boldsymbol{w})$$

for each admissible state–action pair $(s, a) \in S \times A$. Here, $\mathbf{s}$ and $\mathbf{a}$ are binary vectors encoding the state $s$ and action $a$ on the state nodes and action nodes, respectively, of a GBM. In RL, the visible nodes of a GBM are partitioned into two subsets of state nodes $S$ and action nodes $A$. Here, $\boldsymbol{w}$ represents the vector of weights of a GBM as in (3). Each entry $w$ of $\boldsymbol{w}$ can now be trained using the TD(0) update rule:

$$\Delta w^{vh} = \varepsilon(r_n(s_n, a_n) + \gamma Q(s_{n+1}, a_{n+1}) - Q(s_n, a_n)) v \langle h \rangle \quad \text{and} \tag{6}$$

$$\Delta w^{hh'} = \varepsilon(r_n(s_n, a_n) + \gamma Q(s_{n+1}, a_{n+1}) - Q(s_n, a_n)) \langle hh' \rangle , \tag{7}$$

where $\langle h \rangle$ and $\langle hh' \rangle$ are the expected values of the variables and the products of the variables, respectively, in the binary encoding of the hidden nodes with respect to the Boltzmann distribution of the classical Hamiltonian (3).

To develop a FERL method using QBMs, let $\beta = \frac{1}{k_B T}$ be a fixed thermodynamic beta as in the classical case. As before, for an assignment of visible variables $\mathbf{v}$, $F(\mathbf{v})$ denotes the equilibrium free energy, and is given via

$$F(\mathbf{v}) := -\frac{1}{\beta} \ln Z_{\mathbf{v}} = \langle \mathcal{H}_{\mathbf{v}} \rangle + \frac{1}{\beta} \operatorname{tr}(\rho_{\mathbf{v}} \ln \rho_{\mathbf{v}}). \tag{8}$$

Here, $Z_{\mathbf{v}} = \operatorname{tr}(e^{-\beta \mathcal{H}_{\mathbf{v}}})$ is the partition function of the clamped QBM and $\rho_{\mathbf{v}}$ is the density matrix $\rho_{\mathbf{v}} = \frac{1}{Z_{\mathbf{v}}} e^{-\beta \mathcal{H}_{\mathbf{v}}}$. The term $-\operatorname{tr}(\rho_{\mathbf{v}} \ln \rho_{\mathbf{v}})$ is the entropy of the system. Note that (8) is a generalization of (5). The notation $\langle \cdots \rangle$ is used for the expected value of any observable with respect to the Gibbs measure (i.e., the Boltzmann distribution), in particular,

$$\langle \mathcal{H}_{\mathbf{v}} \rangle = \frac{1}{Z_{\mathbf{v}}} \operatorname{tr}(\mathcal{H}_{\mathbf{v}} e^{-\beta \mathcal{H}_{\mathbf{v}}}).$$

This is also a generalization of the weighted sum $\sum_{\mathbf{h}} \mathbb{P}(\mathbf{h}|\mathbf{v}) \mathscr{E}_{\mathbf{v}}(\mathbf{h})$ in (5). Inspired by the ideas of Sallans & Hinton (2004) and Amin et al. (2016), we use the negative free energy of a QBM to approximate the $Q$-function exactly as in the classical case:

$$Q(s, a) \approx -F(\mathbf{s}, \mathbf{a}; \boldsymbol{w})$$

for each admissible state–action pair $(s, a) \in S \times A$. As before, $\mathbf{s}$ and $\mathbf{a}$ are binary vectors encoding the state $s$ and action $a$ on the state nodes and action nodes, respectively, of a Boltzmann machine. In RL, the visible nodes of a Boltzmann machine are partitioned into two subsets of state nodes $S$ and action nodes $A$. Here, $\boldsymbol{w}$ represents the vector of weights of a QBM as in (4). Each entry $w$ of $\boldsymbol{w}$ can now be trained using the TD(0) update rule:

$$\Delta w = -\varepsilon(r_n(s_n, a_n) - \gamma F(s_{n+1}, a_{n+1}) + F(s_n, a_n)) \frac{\partial F}{\partial w}.$$

As shown in Crawford et al. (2018), from (8) we obtain

$$\Delta w^{vh} = \varepsilon(r_n(s_n, a_n) \tag{9}$$
$$- \gamma F(s_{n+1}, a_{n+1}) + F(s_n, a_n)) v \langle \sigma_h^z \rangle \quad \text{and}$$

$$\Delta w^{hh'} = \varepsilon(r_n(s_n, a_n) \tag{10}$$
$$- \gamma F(s_{n+1}, a_{n+1}) + F(s_n, a_n)) \langle \sigma_h^z \sigma_{h'}^z \rangle.$$

This concludes the development of the FERL method using QBMs. We refer the reader to Algorithm 3 in Crawford et al. (2018) for more details. What remains to be done is to approximate values of the free energy $F(s, a)$ and also the expected values of the observables $\langle \sigma_h^z \rangle$ and $\langle \sigma_h^z \sigma_{h'}^z \rangle$. In this paper, we demonstrate how quantum annealing can be used to address this challenge.

## 2.3 Adiabatic evolution of open quantum systems

The evolution of a quantum system under a slowly changing, time-dependent Hamiltonian is characterized by Born & Fock (1928). The *quantum adiabatic theorem* (QAT) in Born & Fock (1928) states that a system remains in its instantaneous steady state, provided there is a gap between the eigen-energy of the steady state and the rest of the Hamiltonian's spectrum at every point in time. QAT motivated Farhi et al. (2000) to introduce a paradigm of quantum computing known as quantum adiabatic computation which is closely related to the quantum analogue of simulated annealing, namely *quantum annealing* (QA), introduced by Kadowaki & Nishimori (1998).

The history of QA and QAT inspired manufacturing efforts towards physical realizations of adiabatic evolution via quantum hardware Johnson et al. (2011). In reality, the manufactured chips are operated at a non-zero temperature and are not isolated from their environment. Therefore, the existing adiabatic theory does not cover the behaviour of these machines. A contemporary investigation in quantum adiabatic theory was therefore initiated to study adiabaticity in open quantum systems Sarandy & Lidar (2005); Venuti et al. (2016); Albash et al. (2012); Avron et al. (2012); Bachmann et al. (2016). These sources prove adiabatic theorems for open quantum systems under various assumptions, in particular when the quantum system is coupled to a thermal bath satisfying the Kubo–Martin–Schwinger condition, implying that the instantaneous steady state is the instantaneous Gibbs state. This work in progress shows promising opportunities to use quantum annealers to sample from the Gibbs state of a TFIM.

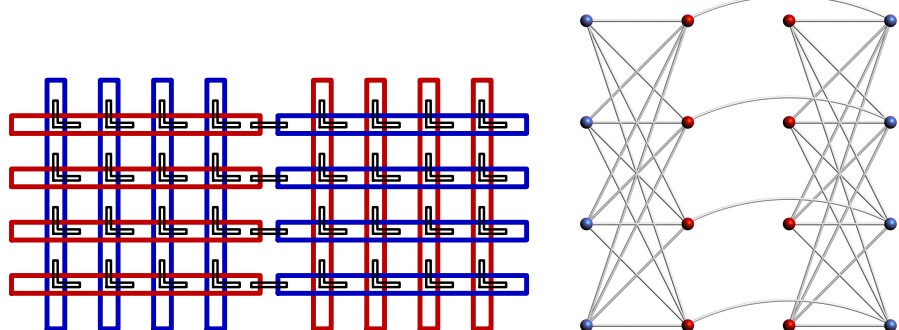

Figure 1: (left) Two adjacent unit cells of the D-Wave 2000Q chip. The intra-cell couplings provide a fully connected bipartite subgraph. However, there are only four inter-cell couplings. (right) The Chimera graph representing the connectivity of the two unit cells of qubits.

In practice, due to additional complications (e.g., level crossings and gap closure, described in the references above), the samples gathered from the quantum annealer are far from the Gibbs state of the final Hamiltonian. In fact, Amin (2015) suggests that the distribution of the samples would instead correspond to an instantaneous Hamiltonian at an intermediate point in time, called the *freeze-out point*. Unfortunately, this point and, consequently, the strength $\Gamma$ of the transverse field at this point, is not known a priori, and also depends on the TFIM undergoing evolution. Our goal is simply to associate a single (average) *virual* $\Gamma$ to all TFIMs constructed through FERL. Another unknown parameter is the inverse temperature $\beta$, at which the Gibbs state, the partition function, and the free energy are attained. In a similar fashion, we wish to associate a single *virtual* $\beta$ to all TFIMs encountered.

The quantum annealer used in our experiments is the D-Wave 2000Q, which consists of a chip of superconducting qubits connected to each other according to a sparse adjacency graph called the *Chimera graph*. The Chimera graph structure looks significantly different from the frequently used models in machine learning, for example, RBMs and DBMs, which consist of consecutive fully connected bipartite graphs. Fig. 1 shows two adjacent blocks of the Chimera graph which consist of 16 qubits, which, in this paper, serve as the clamped QBM used in FERL.

Another complication when using a quantum annealer as a QBM is that the spin configurations of the qubits can only be measured along a fixed axis (here the $z$-basis of the Bloch sphere). Once $\sigma^z$ is measured, all of the quantum information related to the projection of the spin along the transverse field (i.e., the spin $\sigma^x$) collapses and cannot be retrieved. Therefore, even with a choice of virtual $\Gamma$, virtual $\beta$, and all of the measured configurations, the energy of the TFIM is still unknown. We propose a method for overcoming this challenge based on the Suzuki–Trotter expansion of the TFIM, which we call *replica stacking*, the details of which are explained in §3.4. In §4, we perform a grid search over values of the virtual parameters $\beta$ and $\Gamma$. The accepted virtual parameters are the ones that result in the most-effective learning for FERL in the early stages of training.

## 3 FREE ENERGY OF QUANTUM BOLTZMANN MACHINES

### 3.1 SUZUKI–TROTTER REPRESENTATION

By the Suzuki–Trotter decomposition Suzuki (1976), the partition function of the TFIM defined by the Hamiltonian (4) can be approximated using the partition function of a classical Hamiltonian denoted by $\mathcal{H}_{\mathbf{v}}^{\text{eff}}$ and called an effective Hamiltonian, which corresponds to a classical Ising model of one dimension higher. More precisely,

$$
\mathcal{H}_{\mathbf{v}}^{\text{eff}}(\mathbf{h}) = -\sum_{\{h,h'\}\subseteq H}\sum_{k=1}^{r}\frac{w^{hh'}}{r}h_k h_k' - \sum_{v,h}\sum_{k=1}^{r}\frac{w^{vh}v}{r}h_k \tag{11}
$$
$$
- w^+\left(\sum_h\sum_{k=1}^{r}h_k h_{k+1} + \sum_h h_1 h_r\right),
$$

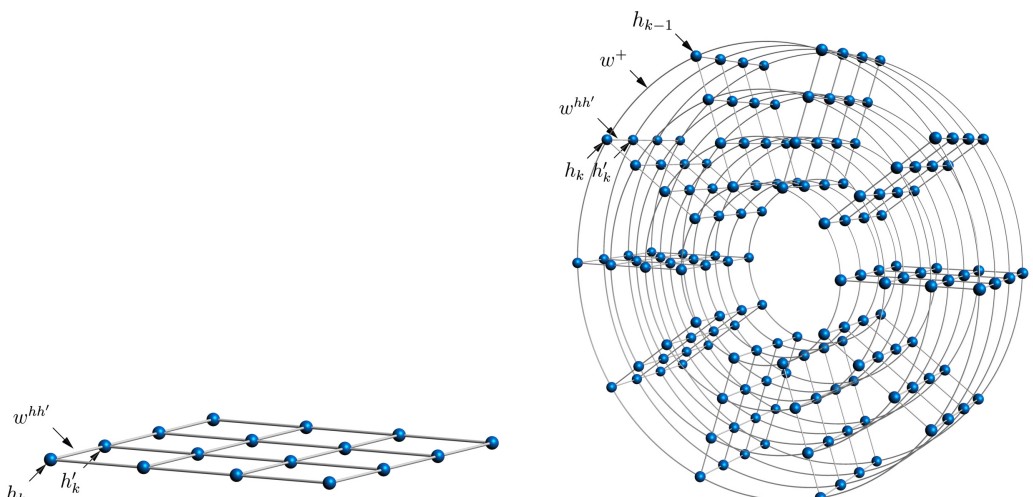

Figure 2: (left) A TFIM consisting of 16 qubits arranged on a two-dimensional lattice with nearest-neighbour couplings. (right) The corresponding effective classical Ising model with ten replicas arranged in a three-dimensional solid torus.

where $r$ is the number of replicas, $w^+ = \frac{1}{2\beta} \log \coth\left(\frac{\Gamma\beta}{r}\right)$, and $h_k$ represent spins of the classical system of one dimension higher. Note that each hidden node's Pauli $z$-matrices $\sigma_h^z$ are represented by $r$ classical spins, denoted by $h_k$, with a slight abuse of notation. In other words, the original Ising model with a non-zero transverse field represented through non-commuting operators can be mapped to a classical Ising model of one dimension higher. Fig. 2 shows the underlying graph of a TFIM on a two-dimensional lattice and a corresponding 10-replica effective Hamiltonian in three dimensions.

The intuition behind the Suzuki–Trotter decomposition is that the superposition of the spins in a quantum system is represented classically by replicas in the $z$-basis. In other words, the measurement of the quantum system in the $z$-basis is interpreted as choosing one replica at random. Note that the probabilities of measuring $+1$ or $-1$ for each individual spin are preserved. This way, each hidden node in the quantum Boltzmann machine carries more information than a classical one; in fact, a classical representation of this system requires $r$ classical binary units via the Suzuki–Trotter decomposition. Consequently, the connections between the hidden nodes become more complicated in the quantum case as well and can carry more information on the correlations between the hidden nodes. Note that the coupling strengths between the replicas are not arbitrary, but come from the mathematical decomposition following the Suzuki–Trotter formula. As a result, the quantum Boltzmann machine can be viewed as an undirected graphical model but in one dimension higher than the classical Boltzmann machine.

## 3.2 Approximation of free energy using Gibbs sampling

In the case of classical GBMs without further restrictions on the graph structure, $\langle h \rangle$, $\langle hh' \rangle$, and $Q(s, a) \approx -F(\mathbf{s}, \mathbf{a}; \boldsymbol{w})$ are not tractable. Consequently, to perform the weight update in (6) one requires samples from the Boltzmann distribution corresponding to energy function (3) to estimate $\langle h \rangle$, $\langle hh' \rangle$, and $F(\mathbf{s}, \mathbf{a}; \boldsymbol{w})$ empirically. To approximate the right-hand side of (9) and (10), we sample from the Boltzmann distribution of the energy function represented by the effective Hamiltonian using (Suzuki, 1976, Theorem 6). We find the expected values of the observables $\langle \sigma_h^z \rangle$ and $\langle \sigma_h^z \sigma_{h'}^z \rangle$ by averaging the corresponding classical spin values. To approximate the free energy of a QBM and

consequently a $Q$-function, we use (Suzuki, 1976, Theorem 4) to substitute (8) by

$$
\begin{aligned}
F(\mathbf{v}) &= \langle \mathcal{H}_{\mathbf{v}}^{\text{eff}} \rangle + \frac{1}{\beta} \sum_{c_{\text{eff}}} \mathbb{P}(c_{\text{eff}}|\mathbf{v}) \log \mathbb{P}(c_{\text{eff}}|\mathbf{v}) \\
&= - \sum_{\{h,h'\}\subseteq H} \sum_{k=1}^{r} \frac{w^{hh'}}{r} \langle h_k h'_k \rangle - \sum_{v,h} \sum_{k=1}^{r} \frac{w^{vh}v}{r} \langle h_k \rangle \\
&\quad - w^{+} \left( \sum_{h} \sum_{k=1}^{r} \langle h_k h_{k+1} \rangle + \sum_{h} \langle h_1 h_r \rangle \right) + \frac{1}{\beta} \sum_{c_{\text{eff}}} \mathbb{P}(c_{\text{eff}}|\mathbf{s},\mathbf{a}) \log \mathbb{P}(c_{\text{eff}}|\mathbf{s},\mathbf{a}),
\end{aligned}
\tag{12}
$$

where $\mathcal{H}_{\mathbf{v}}^{\text{eff}}$ is the effective Hamiltonian and $c_{\text{eff}}$ ranges over all spin configurations of the effective classical Ising model of one dimension higher, defined by $\mathcal{H}_{\mathbf{v}}^{\text{eff}}$. Here, $\langle h \rangle$ and $\langle hh' \rangle$ are the expected values of the variables and the products of the binary variables, respectively, with respect to the Boltzmann distribution of the classical effective Hamiltonian (11).

### 3.3 SIMULATED QUANTUM ANNEALING

One way to sample spin values from the Boltzmann distribution of the effective Hamiltonian is to use the simulated quantum annealing algorithm (SQA) (see (Brabazon et al., 2015, p. 422) for an introduction). SQA is one of the many flavours of quantum Monte Carlo methods, and is based on the Suzuki–Trotter expansion described above. This algorithm simulates the quantum annealing phenomena of a TFIM by slowly reducing the strength of the transverse field at finite temperature to the desired target value. In our implementation, we have used a single spin-flip variant of SQA with a linear transverse-field schedule as in Martoňák et al. (2002) and Heim et al. (2015). Experimental studies have shown similarities in the behaviour of SQA and that of quantum annealing Isakov et al. (2015); Albash et al. (2014) and its physical realization by D-Wave Systems Brady & van Dam (2016); Shin et al. (2014).

The classical counterpart of SQA is conventional simulated annealing (SA), which is based on thermal annealing. This algorithm can be used to sample from Boltzmann distributions that correspond to an Ising spin model in the absence of a transverse field. Unlike SA, it is possible to use SQA not only to approximate the Boltzmann distribution of a classical Boltzmann machine, but also that of a quantum Hamiltonian in the presence of a transverse field. This can be done by reducing the strength of the transverse field to the desired value defined by the model, rather than to zero. It has been proven by Morita & Nishimori (2006) that the spin system defined by SQA converges to the Boltzmann distribution of the effective classical Hamiltonian of one dimension higher that corresponds to the quantum Hamiltonian. Therefore, it is straightforward to use SQA to approximate the free energy in (12) as well as the observables $\langle \sigma_h^z \rangle$ and $\langle \sigma_h^z \sigma_{h'}^z \rangle$. However, any Boltzmann distribution sampling method based on Markov chain Monte Carlo (MCMC) has the major drawback of being extremely slow and computationally involved. Actually, it is an NP-hard problem to sample from the Boltzmann distribution. Another option is to use variational approximation Salakhutdinov & Hinton (2009), which suffers from lack of accuracy and works in practice only in limited cases. As explained above, quantum annealers have the potential to provide samples from Boltzmann distributions (in the $z$-basis) corresponding to TFIM in a more efficient way. In what follows, we explain how to use quantum annealing to approximate the free energy corresponding to an effective Hamiltonian which in turn can be used to approximate the free energy of a QBM.

### 3.4 REPLICA STACKING

As explained in §2.3, a quantum annealer provides measurements of $\sigma^z$ spins for each qubit in the TFIM. The observables $\langle \sigma_h^z \rangle$ and $\langle \sigma_h^z \sigma_{h'}^z \rangle$ can therefore be approximated by averaging over the spin configurations measured by the quantum annealer. Moreover, by (Suzuki, 1976, Theorem 6) and translation invariance, each replica of the effective classical model is an approximation of the spin measurements of the TFIM in the measurement bases $\sigma^z$. Therefore, a $\sigma^z$-configuration sampled by a quantum annealer that operates at a given *virtual* inverse temperature $\beta$, and anneals up to a *virtual* transverse-field strength $\Gamma$, may be viewed as an instance of a classical spin configuration from a replica of the classical effective Hamiltonian of one dimension higher.

This suggests the following method to approximate the free energy from (12) for a TFIM. We gather a pool $\mathscr{C}$ of configurations sampled by the quantum annealer for the TFIM considered, allowing repetitions. Let $r$ be the number of replicas. We write $c_{\text{eff}} = (c_1, \ldots, c_r)$ to indicate an effective configuration $c_{\text{eff}}$ with the classical configurations $c_1$ to $c_r$ as its replicas. We write $\underline{c_{\text{eff}}}$ to denote the underlying set $\{c_1, \ldots, c_r\}$ of replicas of $c_{\text{eff}}$ (without considering their ordering). We have

$$\mathbb{P}\left[c_{\text{eff}} = (c_1, \ldots, c_r)\right]$$
$$= \mathbb{P}\left[c_{\text{eff}} = (c_1, \ldots, c_r)|\underline{c_{\text{eff}}} = \{c_1, \ldots, c_r\}\right] \times \mathbb{P}\left[\underline{c_{\text{eff}}} = \{c_1, \ldots, c_r\}\right]$$
$$= \mathbb{P}\left[c_{\text{eff}} = (c_1, \ldots, c_r)|\underline{c_{\text{eff}}} = \{c_1, \ldots, c_r\}\right] \times \mathbb{P}\left[\underline{c_{\text{eff}}} = \{c_1, \ldots, c_r\}|\underline{c_{\text{eff}}} \subseteq \mathscr{C}\right] \times \mathbb{P}\left[\underline{c_{\text{eff}}} \subseteq \mathscr{C}\right].$$

The argument in the previous paragraph can now be employed to allow the assumption

$$\mathbb{P}\left[\underline{c_{\text{eff}}} \subseteq \mathscr{C}\right] \simeq 1.$$

In other words, the probability mass function of the effective configurations is supported in the subset of those configurations synthesized from the elements of $\mathscr{C}$ as *candidate replicas*.

The conditional probability $\mathbb{P}[\underline{c_{\text{eff}}} = \{c_1, \ldots, c_r\}|\underline{c_{\text{eff}}} \subseteq \mathscr{C}]$ can be sampled from by drawing $r$ elements $c_1, \ldots, c_r$ from $\mathscr{C}$. We then sample from $\mathbb{P}\left[c_{\text{eff}} = (c_1, \ldots, c_r)|\underline{c_{\text{eff}}} = \{c_1, \ldots, c_r\}\right]$, according to the following distribution over $c_{\text{eff}}$:

$$\pi(c_{\text{eff}}) \triangleq \mathbb{P}\left[c_{\text{eff}} = (c_1, \ldots, c_r)|\underline{c_{\text{eff}}} = \{c_1, \ldots, c_r\}\right]$$
$$= \frac{e^{\beta w^+ \sum_h \left(\sum_{k=1}^{r-1} h_{c_k} h_{c_{k+1}} + h_{c_1} h_{c_r}\right)}}{\sum_{\underline{c_{\text{eff}}}=\{c_1, \ldots, c_k\}} e^{\beta w^+ \sum_h \left(\sum_{k=1}^{r-1} h_{c_k} h_{c_{k+1}} + h_{c_1} h_{c_r}\right)}}.$$

We consider $\pi(c_{\text{eff}})$ our target distribution and construct the following MCMC method for which the limiting distribution is $\pi(c_{\text{eff}})$. We first attach the $r$ classical spin configurations to the SQA's effective configuration structure uniformly at random. We then transition to a different arrangement with a Metropolis acceptance probability. For example, we may choose two classical configurations at random and exchange them with probability

$$p(c_{\text{eff}}, c'_{\text{eff}}) = \min\left\{1, \exp\left(\beta(E(c'_{\text{eff}}) - E(c_{\text{eff}}))\right)\right\}, \tag{13}$$

where $E(c_{\text{eff}}) = w^+ \sum_h \left(\sum_{k=1}^{r-1} h_{c_k} h_{c_{k+1}} + h_{c_1} h_{c_r}\right)$. Such a stochastic process is known to satisfy the detailed balance condition. Consequently, the MCMC method allows us to sample from the effective spin configurations. This procedure of sampling and then performing the MCMC method creates a pool of effective spin configurations, which are then employed in equation (12) in order to approximate the free energy of the TFIM empirically.

However, we consider a relatively small number of hidden nodes in our experiments, so the number of different $\sigma^z$-configurations sampled by the quantum annealer is limited. As a consequence, there is no practical need to perform the MCMC method defined above. Instead, we attach classical spin configurations from the pool to the SQA effective configuration structure at random. In other words, in $r$ iterations, a spin configuration is sampled from the pool of classical spin configurations described above and inserted as the next replica of the effective classical Hamiltonian consisting of $r$ replicas.

It is worthwhile to reiterate that this replica stacking technique yields an undirected graphical model. Specifically, the structure described in Fig. 2 (right) is an undirected graphical model in the space of hidden nodes, where the node statistics are obtained from the Boltzmann distribution. One difference between this model and a classical Boltzmann machine is that each hidden node activation is governed by a series of $r$ replicas in one dimension higher, and the undirected, replica-to-replica connections calculated therein. Moreover, the energy function of this extended model differs from the energy function of the classical Boltzmann machine (compare (11) and (3)). The free energy of the extended graphical model serves as the function approximator to the $Q$-function.

## 4 THE EXPERIMENTS

We benchmark our various FERL methods on a $3 \times 5$ grid-world problem Sutton (1990) with an agent capable of taking the actions up, down, left, or right, or standing still, on a grid-world with

---

**Algorithm 1** FERL-QBM

---
    initialize weights of QBM
    **for all** training samples $(s_1, a_1)$ **do**
        $s_2 \leftarrow a_1(s_1)$, $a_2 \leftarrow \text{argmax}_a Q(s_2, a)$
        calculate $\langle \sigma^z_{h_i} \rangle, \langle \sigma^z_{h_i} \sigma^z_{h'_i} \rangle, \langle \mathcal{H}^{\text{eff}}_{\mathbf{s_i},\mathbf{a_i}} \rangle, and\, \mathbb{P}(c_{eff}|\mathbf{s_i}, \mathbf{a_i})$ using Algorithm 2, for $(i = 1, 2)$
        calculate $F(\mathbf{s}_i, \mathbf{a}_i)$ using (12) for $(i = 1, 2)$
        $Q(s_i, a_i) \leftarrow -F(\mathbf{s}_i, \mathbf{a}_i)$ for $(i = 1, 2)$
        update QBM weights using (9) and (10)
        $\pi(s_1) \leftarrow \text{argmax}_a Q(s_1, a)$
    **end for**
    **return** $\pi$

---

**Algorithm 2** Replica stacking

---
    initialize the structure of the effective Hamiltonian in one dimension higher
    **for** $i = 1, 2, ..., m$ **do**
        **for** $j = 1, 2, ..., r$ **do**
            obtain spin configuration sample in $z$-basis from QA
            attach this spin configuration to $j$-th replica of the $i$-th effective configuration structure
        **end for**
        perform the MCMC technique described in §3.4 with transition probabilities (13) to obtain the $i$-th instance of effective spin configurations
    **end for**
    obtain $\langle \mathcal{H}^{\text{eff}}_{\mathbf{s_i},\mathbf{a_i}} \rangle$ from the average energy of the $m$ effective spin configurations
    obtain $\langle h \rangle$ and $\langle hh' \rangle$ by averaging over all $h$ and $h'$ replicas in each spin configuration
    gather statistics from $\mathbb{P}(c_{\text{eff}}|\mathbf{s_i}, \mathbf{a_i})$ using the $m$ effective spin configurations
    **return** $\langle h \rangle, \langle hh' \rangle, \langle \mathcal{H}^{\text{eff}}_{\mathbf{s_i},\mathbf{a_i}} \rangle$, and $\mathbb{P}(c_{\text{eff}}|\mathbf{s_i}, \mathbf{a_i})$

---

one deterministic reward, one wall, and one penalty, as shown in Fig. 3 (top). The task is to find an optimal policy, as shown in Fig. 3 (bottom), for the agent at each state in the grid-world. All of the Boltzmann machines used in our algorithms consist of 16 hidden nodes.

The discount factor, as explained in §2, is set to $0.8$. The agent attains the reward $R = 200$ in the top-left corner, the neutral value of moving to any empty cell is 100, and the agent is penalized by not receiving any reward if it moves to the penalty cell with value $P = 0$.

For $T_r$ independent runs of every FERL method, $T_s$ training samples are used. The fidelity measure at the $i$-th training sample is defined by

$$\text{fidelity}(i) = (T_r \times |S|)^{-1} \sum_{l=1}^{T_r} \sum_{s \in S} \mathbb{1}_{A(s,i,l) \in \pi^*(s)}, \tag{14}$$

where $\pi^*$ denotes the best known policy and $A(s, i, l)$ denotes the action assigned at the $l$-th run and $i$-th training sample to the state $s$. In our experiments, each algorithm is run 100 times.

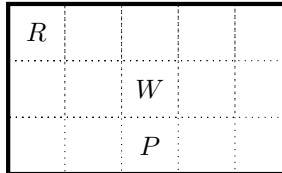 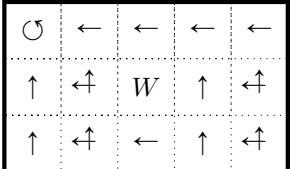

Figure 3: (left) A $3 \times 5$ grid-world problem instance with one reward, one wall, and one penalty. (right) An optimal policy for this problem instance can be represented as a selection of directional arrows indicating movement directions.

Fig. 4 demonstrates the performance of a fully connected deep $Q$-network Mnih et al. (2015) consisting of an input layer of 14 state nodes, two layers of eight hidden nodes each, and an output layer of five nodes representing the values of the $Q$-function for different actions, given a configuration of state nodes. We use the same number of hidden nodes in the fully connected deep $Q$-network as in the other networks described in this paper.

## 4.1 GRID SEARCH FOR VIRTUAL PARAMETERS ON THE D-WAVE 2000Q

We treat the network of superconducting qubits represented in Fig. 1 as a clamped QBM with two hidden layers, represented using blue and red colours. The state nodes are considered fully connected to the blue qubits and the action nodes are fully connected to the red qubits.

For a choice of virtual parameters $\Gamma \neq 0$ and $\beta$, which appear in (11) and (12), and for each query to the D-Wave 2000Q chip, we construct 150 effective classical configurations of one dimension higher, out of a pool of 3750 reads, according to the replica stacking method introduced in §3.4. The 150 configurations are, in turn, employed to approximate the free energy of the quantum Hamiltonian. We conduct 10 independent runs of FERL in this fashion, and find the average fidelity over the 10 runs and over the $T_s = 300$ training samples.

Fig. 5 shows a heatmap of the average fidelity of each choice of virtual parameters $\beta$ and $\Gamma$. In the $\Gamma = 0$ row, each query to the D-Wave 2000Q is considered to be sampling from a classical GBM with Fig. 1 as the underlying graph.

## 4.2 FERL FOR THE GRID-WORLD PROBLEM

Fig. 6 shows the growth of the average fidelity of the best known policies generated by different FERL methods. For each method, the fidelity curve is an average over 100 independent runs, each with $T_s = 500$ training samples.

In this figure, the "D-Wave $\Gamma = 0.5$, $\beta = 2.0$" curve corresponds to the D-Wave 2000Q replica stacking-based method with the choice of the best virtual parameters $\Gamma = 0.5$ and $\beta = 2.0$, as shown in the heatmap in Fig. 5. The training is based on formulae (9), (10), and (12). The "SQA Bipartite $\Gamma = 0.5$, $\beta = 2.0$" and "SQA Chimera $\Gamma = 0.5$, $\beta = 2.0$" curves are based on the same formulae with the underlying graphs being a bipartite (DBM) and a Chimera graph, respectively, with the same choice of virtual parameters, but the effective Hamiltonian configurations generated using SQA as explained in §3.3.

The "SA Bipartite $\beta = 2.0$" and "SA Chimera $\beta = 2.0$" curves are generated by using SA to train a classical DBM and a classical GBM on the Chimera graph, respectively, using formulae (6), (7), and (5). SA is run with a linear inverse temperature schedule, where $\beta = 2.0$ indicates the final value. The "D-Wave Classical $\beta = 2.0$" curve is generated using the same method, but with samples

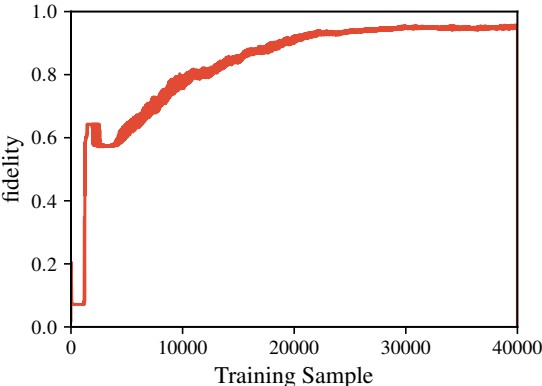

Figure 4: The learning curve of a fully connected deep $Q$-network with two hidden layers, each with eight hidden nodes, for the grid-world problem instance shown in Fig. 3.

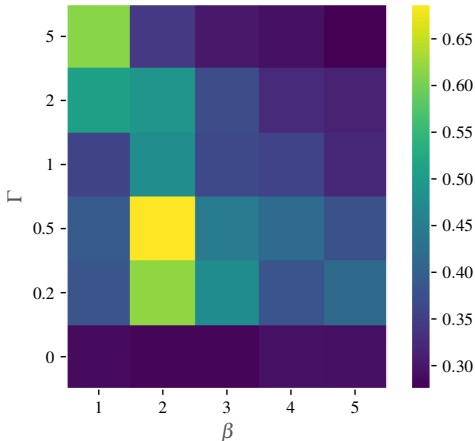

Figure 5: Heatmap of average fidelity observed using various choices of virtual parameters $\beta$ and $\Gamma$. The $\Gamma = 0$ row tests the performance of FERL with samples obtained from the quantum annealer treated as classical configurations of a GBM. In all other rows, samples are interpreted as $\sigma^z$-measurements of a QBM.

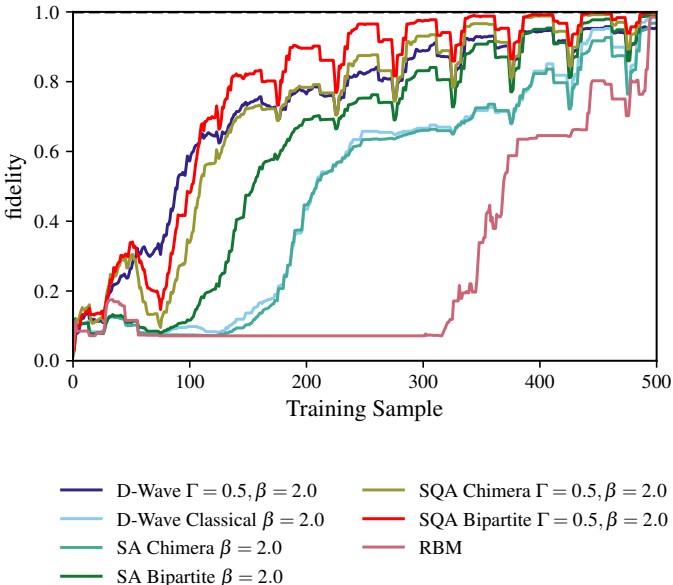

Figure 6: Comparison of different FERL methods for the grid-world problem instance in Fig. 3.

obtained using the D-Wave 2000Q. The "RBM" curve is generated using the method in Sallans & Hinton (2004).

## 5 DISCUSSION

We solve the grid-world problem using various $Q$-learning methods with the $Q$-function parametrized by different neural networks. For comparison, we demonstrate the performance of a fully connected deep $Q$-network method that can be considered state of the art. This method efficiently processes every training sample, but, as shown in Fig. 4, requires a very large number of training samples to converge to the optimal policy. Another conventional method is free-energy-based RL using an RBM. This method is also very successful at learning the optimal policy at the scale of the RL task

considered in our experiment. Although this method does not outperform other FERL methods that take advantage of a highly efficient sampling oracle, the processing of each training sample is efficient, as it is based on closed formulae. In fact, for the size of problem considered, the RBM-based FERL outperforms the fully connected deep $Q$-network method.

The comparison of results in Fig. 6 suggests that replica stacking is a successful method for estimating effective classical configurations obtained from a quantum annealer, given that the spins can only be measured in measurement bases. For practical use in RL, this method provides a means of treating the quantum annealer as a QBM. FERL using the quantum annealer, in conjunction with the replica stacking technique, provides significant improvement over FERL using classical Boltzmann machines. The curve representing SQA-based FERL using a Boltzmann machine on the Chimera graph is almost coincident with the one obtained using the D-Wave 2000Q, whereas the SQA-based FERL using a DBM slightly outperforms it. This suggests that quantum annealing chips with greater connectivity and more control over annealing time can further improve the performance of the replica stacking method applied to RL tasks. This is further supported by comparing the performance of SA-based FERL using a DBM versus SA-based FERL using the Chimera graph. This result shows that DBM is, due to its additional connections, a better choice of neural network compared to the Chimera graph.

For practical reasons, we aim to associate an identical choice of virtual parameters $\beta$ and $\Gamma$ to all of the TFIMs constructed using FERL. Benedetti et al. (2016) and Raymond et al. (2016) provide methods for estimating the effective inverse temperature $\beta$ for other applications. However, in both studies, the samples obtained from the quantum annealer are matched to the Boltzmann distribution of a classical Ising model. In fact, the transverse-field strength is a second virtual parameter that we consider. The optimal choice $\Gamma = 0.5$ corresponds to $2/3$ of the annealing time, in agreement with the work of Amin (2015), who also considers TFIM with 16 qubits.

The agreement of FERL using quantum annealer reads treated as classical Boltzmann samples with that of FERL using SA and classical Boltzmann machines suggests that, at least for this task and this size of Boltzmann machine, the measurements provided by the D-Wave 2000Q can be considered good approximations of Boltzmann distribution samples of classical Ising models.

The extended undirected graphical model developed in this paper using the replica stacking method is not limited to $Q$-function approximation in RL tasks. Potentially, this method can be applied to tasks where Boltzmann machines can be used. This method provides a mechanism for approximating the activations and partition functions of quantum Boltzmann machines that have a significant transverse field.

## 6 CONCLUSION

In this paper, we describe a free-energy-based reinforcement learning algorithm using an existing quantum annealer, namely the D-Wave 2000Q. Our method relies on the Suzuki–Trotter decomposition and the use of the measured configurations by the D-Wave 2000Q as replicas of an effective classical Ising model of one dimension higher. The results presented here are first-step proofs of concept of a proposed quantum algorithm with a promising path towards outperforming reinforcement learning algorithms devised for digital hardware. Given appropriate advances in quantum annealing hardware, future research can employ the proposed principles to solve larger-scale reinforcement learning tasks in the emerging field of quantum machine learning.

ACKNOWLEDGEMENTS

The authors would like to thank Marko Bucyk for editing this manuscript.

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
