# OpenReview forum: "Reinforcement Learning via Replica Stacking of Quantum Measurements for the Training of Quantum Boltzmann Machines"
_ICLR.cc/2018/Conference — Reject_

### Official Review · AnonReviewer2 · 2017-11-23

**Rating:** 4
**Confidence:** 3

**Review:**

Summary: The paper demonstrates the use of a quantum annealing machine to solve a free-energy based reinforcement learning problem. Experimental results are demonstrated on a toy gridworld task, where if I understand correctly it does better than a DQN and a method based on RBM-free-energy approximation (Sallans and Hinton, 2004)

Clarity: The paper is very hard to read. It seems to be targeted towards a physics/quantum hardware crowd rather than a machine learning audience. I think most readers, even those very familiar with probabilistic models and RL, would find reading the paper difficult due to jargon/terminology and poorly explained concepts. The paper would need a major rewrite to be of interest to the ML community.

Relevance: RL, probabilistic models, and function approximators are all relevant topics. However, the focus of the paper seems to be on parts (like hardware aspects) that are not particularly relevant to the ML community. I have a hard time imagining follow-up work on this, given that the experiments are run on a toy task and require specialized hardware (so they would be extremely difficult to reproduce/improve upon).

Soundness: I can't judge the technical soundness as it is mostly outside my expertise. I wonder if the main algorithm the work is based on (Crawford et al, 2016) has been peer-reviewed (citation appears to be a preprint, and couldn't find a conference/journal version).

Significance: It's good to know that the the quantum annealing machine can be used for RL. However, the method of choice (free-energy based Q-function approximation) seems a bit exotic, and the experimental results are extremely underwhelming (5x3 gridworld).

---

> ### Author Response · Authors · 2018-01-05
> **Thank you very much for the comments and suggestions!**
>
> "Clarity: The paper is very hard to read. It seems to be targeted towards a physics/quantum hardware crowd rather than a machine learning audience. I think most readers, even those very familiar with probabilistic models and RL, would find reading the paper difficult due to jargon/terminology and poorly explained concepts. The paper would need a major rewrite to be of interest to the ML community."
>
> - We definitely want to make our work accessible to the ML community so we find your guidance extremely helpful. We have added more details about the algorithm. We added further explanations on classical and quantum Boltzmann machines, the similarities and differences between the two, and how the quantum BMs generalize their classical counterparts. We have used and referred to the terminology of  “Reinforcement Learning with Factored States and Actions” by Brian Sallans and Geoffrey E. Hinton as much as possible for further clarity.
>
> "Relevance: RL, probabilistic models, and function approximators are all relevant topics. However, the focus of the paper seems to be on parts (like hardware aspects) that are not particularly relevant to the ML community. I have a hard time imagining follow-up work on this, given that the experiments are run on a toy task and require specialized hardware (so they would be extremely difficult to reproduce/improve upon)."
>
> - We hope that our work serves a small part in setting the ground foundations of new trends in applicability of quantum and exotic hardware in solving real-world machine learning problems. We foresee that a collaboration between the physics and ML community can be crucial to achieving this goal. The quantum computing community is working diligently on making the prototype hardwares accessible through cloud services and to provide user-friendly packages and APIs that hide the cumbersome details and encourage engagement of researchers from other disciplines.
>
> "Soundness: I can't judge the technical soundness as it is mostly outside my expertise. I wonder if the main algorithm the work is based on (Crawford et al, 2016) has been peer-reviewed (citation appears to be a preprint, and couldn't find a conference/journal version)."
>
> - Crawford et al (2016) is peer reviewed, being accepted for publication in Quantum Information & Computation (early 2018), presented as a talk at Theory of Quantum Computation (2017), and presented as a poster at Adiabatic Quantum Computation (2017).
>
> "Significance: It's good to know that the the quantum annealing machine can be used for RL. However, the method of choice (free-energy based Q-function approximation) seems a bit exotic, and the experimental results are extremely underwhelming (5x3 gridworld)."
>
> - Free-energy based Q-function approximation is an exotic choice for classical computations but a natural choice in the context of Quantum Annealers. We agree that the experimental results are underwhelming due to limitations to the existing prototype hardware and the computationally expensive classical simulations. Moving to larger and more useful benchmarks is definitely the most important future direction to this work.
>
> Thank you again for your helpful review, we hope that we have been able to address your concerns.

---

### Official Review · AnonReviewer3 · 2017-11-26
**Interesting but maybe far from the main topic of the conference**

**Rating:** 6
**Confidence:** 4

**Review:**

The paper is easy to read  for a physicist, but I am not sure how useful it would be for ICLR... it is not clear for me it there is an interest for quantum problems in this conference. This is something I will let to the Area Chair to deceede. Other than this, the paper is interesting, certainly correct, and provides a nice perspective on the future of learning with quantum computers. I like the  quantum "boltzmann machine" problems.

I feel, however, but it might be a bit far from the main interest of the conference.

Comments:

* What the authors called "Free energy-based reinforcement learning" seems to me just the minimization / maximiation of the free energy. This is simply maximum likelihood applied to the free energy and I think that calling it "reinforcement learning" is not only wrong, but also is very confusing, given this is usually reserved to an entirely different learning process.

* While i liked the introduction of the quantum Boltzmann machine, I would be happy to learn what they can do? Are these useful, for instance, to study correlated fermions/bosons? The paper does not explain why one should be concerns with these devices.

* The fact that the simulation on a classical computer agrees with the one on a quantum computer is promising, but I would say that this shows that, so far, there is not yet a clear advantage in using a quantum computer. This might change, but in the mean time, what is the benefits for the ICLR community?

---

> ### Author Response · Authors · 2018-01-05
> **Thank you very much for the positive feedback!**
>
> Although we work in a tangential field to that of machine learning we are excited to bring the progress in quantum machine learning to the attention of the ML experts in the ICLR community. The potential collaboration the can result from such exposition would be invaluable to investigating applicability of quantum computing in real-world computational tasks.
>
> "What the authors called "Free energy-based reinforcement learning" seems to me just the minimization / maximiation of the free energy. This is simply maximum likelihood applied to the free energy and I think that calling it "reinforcement learning" is not only wrong, but also is very confusing, given this is usually reserved to an entirely different learning process. "
>
> - We in fact solve Bellman’s optimality equation rather than minimization/maximization of the free energy. The temporal difference is what is minimized rather than maximum likelihood. The roll of free-energy is to act as a function approximation for the Q-function. Please also refer to “Reinforcement Learning with Factored States and Actions” by Brian Sallans and Geoffrey E. Hinton. We would love to hear your feedback if this clarifies what free-energy based reinforcement learning is meant here. Do you mean that the core idea of replica stacking can be applied not only to Reinforcement Learning but to other machine learning paradigms?
>
> "While i liked the introduction of the quantum Boltzmann machine, I would be happy to learn what they can do? Are these useful, for instance, to study correlated fermions/bosons? The paper does not explain why one should be concerns with these devices."
>
> - The quantum Boltzmann machines are generalization of classical Boltzmann machines, so, in theory, QBMs can replace BMs in all the tasks in which BMs are used. However, the details of how to obtain the samples and other statistics required for the specific task need to be investigated since preparing and reading classical (digital) data from quantum systems is not straightforward: any measurement of a quantum system results collapse of the wavefunction which consequently "stops" the quantum algorithm. In this paper we work out such details for our specific task (i.e. sampling from the transverse-field Ising model).
>
> " The fact that the simulation on a classical computer agrees with the one on a quantum computer is promising, but I would say that this shows that, so far, there is not yet a clear advantage in using a quantum computer. This might change, but in the mean time, what is the benefits for the ICLR community?"
>
> - You’re definitely right that the agreement between the classical simulation and quantum experiment is a verifying step for the proposal here. However, the classical simulations are extremely slow and that is where the advantage in using a quantum computer may lie. Furthermore, even using Monte-Carlo methods only special quantum systems can be simulated classically (this is known as the sign-problem; See for instance arXiv:cond-mat/0408370.) For specific systems without sign problem, Monte-Carlo methods could be viable simulation methods if implemented on specialized hardware (e.g. ASICS). This can potentially result "quantum-inpsired" algorithms that run on classical hardware but take advantage of the extended representational power of the quantum model.
>
> Thank you again for taking the time to review our work, your comments have given us a chance to clarify the intent.

---

### Official Review · AnonReviewer1 · 2017-11-27
**replica stacking for quantum annealers used for RL**

**Rating:** 4
**Confidence:** 3

**Review:**

There is no scientific consensus on whether quantum annealers such as the D-Wave 2000Q that use the transverse-field Ising models yield any gains over classical methods (c.f. https://arxiv.org/abs/1703.00622). However, it is an exciting research area and this paper is an interesting demonstration of the feasibility of using quantum annealers for reinforcement learning.

This paper builds on Crawford et al. (2016), an unpublished preprint, who develop a quantum Boltzmann machine reinforcement learning algorithm (QBM-RL). A QBM consists of adding a transverse field term to the RBM Hamiltonian (negative log likelihood), but the benefits of this for unsupervised tasks are unclear (c.f. https://arxiv.org/abs/1601.02036, another unpublished preprint). QBM-RL consists of using a QBM to model the state-action variables: it is an undirected graphical model whose visible nodes are clamped to observed state-action pairs. The hidden nodes model dependencies between states and actions, and the weights of the model are updated to maximize the free energy or Q function (value of the state-action pair).

The authors extend QBM-RL to work with quantum annealers such as the D-Wave 2000Q, which has a specific bipartite graph structure and requires special consideration because it can only yield samples of hidden variables in a fixed basis. To overcome this, the authors develop a Suzuki-Trotter expansion and call it 'replica stacking', where a classical Hamiltonian in one dimension higher is used to approximate the quantum Hamiltonian. This enables the use of quantum annealers. The authors compare their method to standard baselines in a grid world environment.

Overall, I do not want to criticize the work. It is an interesting proof of concept. But given the high price of quantum annealers, limited applicability of the technique, and unclear benefits of the authors' method, I do not think it is relevant to this specific conference. It may be better suited to a workshop specific to quantum machine learning methods.
=======================================
+ please add an algorithm box for your method. It deviates significantly from QBM-RL. For example, something like: (1) init weights of boltzmann machine randomly (2) sample c_eff ~ C from the pool of configurations sampled from the transverse-field Ising model using a quantum annealer with chimera graph (3) using the samples, calculate effective classical hamiltonian used to approximate the quantum system (4) use the weight update rules derived from Bellman equations (spell out the rules).

+ moving the details of sampling into the appendix would help; they are not important for understanding the main ingredients of your method

There are so many moving parts in your system, and someone without a physics background will struggle to understand it. Clarifying the algorithm in terms familiar to machine learning researchers will go a long way toward helping people understand your method.

+ the benefits of your method is unclear - it looks like the method works, but doesn't outperform the others. this is fine, but it is better to be straightforward about this and bill it as a 'proof of concept'

+ perhaps consider rebranding the paper as something like 'RL using replica stacking for sampling from quantum boltzmann machines with quantum annealers'. Elucidating why replica stacking is a crucial contribution of your work would be helpful, and could be of broad interest in the machine learning community. Right now it is too dense to be useful for the average person without a physics background: what difficulties are intrinsic to a quantum Hamiltonian? What is the intuition behind the Suzuki-Trotter decomposition you develop? What is the 'quantum' Boltzmann machine in machine learning terms (hidden-hidden connections in an undirected graphical model!)? What is replica-stacking in graphical model terms (this would be a great ML contribution in its own right!)? Really spelling these things out in detail (or in the appendix) would help
==========================================
1) eq 14 is malformed

2) references are not well-formatted

3) need factor of 1/2 to avoid double counting in sums over nearest neighbors (please be precise)

---

> ### Author Response · Authors · 2018-01-05
> **Thank you very much for your excellent comments and suggestions!**
>
> In response to your very helpful comments and suggestions, we provide a list of responses:
>
> - Crawford et al (2016) is peer reviewed, published in Quantum Information & Computation QIC Vol.18 No1&2, we have updated this reference in the bibliography to reflect this publication. This paper was also accepted and presented as a talk at the `Theory of Quantum Computation 2017’, and accepted and presented as a poster at `Adiabatic Quantum Computation 2017’.
>
> - On our toy model, our method outperforms the classical approaches investigated in this paper: the RMB-based method and the deep Q networks. It is worth clarifying that in Fig 6. SA Bipartite, SA Chimera, SQA Bipartite and SQA Chimera labels indicate `simulators’ of the behaviour of the quantum device and are not efficient as classical methods. The RBM curve in this figure is the only classical method. Fig 4 demonstrates the performance of the deep Q network which is another classical method considered for comparison. We agree that our experiments do not prove superiority of the proposed  algorithm, as these benchmarks are on a small grid-world problem used to demonstrate a proof of concept. We have added addition comments into the body of the paper to emphasize this point.
>
> =======================================
>
> - We have added two algorithms, one for the FERL-QBM method, another explaining the replica stacking algorithm in detail.
>
> Thank you for all the useful suggestions! We hope to have been able to address all of them by revisions to the paper, including the title change you advised.  We have also edited our paper to reflect the technical formatting issues in formulas and references. With respect to the missing 1/2 factor to avoid double counting in sums over nearest neighbours, we have clarified that the summations run on choices of pairs to avoid possible confusions.

---

### Author Response · Authors · 2018-01-05
**We have made a few changes to the paper in order to make our work more accessible to the ML community.**

We have added more details about our replica stacking algorithm. We have added further explanations on classical and quantum Boltzmann machines, the similarities and differences between the two, and how the quantum Boltzmann machines generalize their classical counterparts. We have used and referred to the terminology of  “Reinforcement Learning with Factored States and Actions” by Brian Sallans and Geoffrey E. Hinton as much as possible for further clarity.

---

### Decision · Program_Chairs · 2018-01-29
**ICLR 2018 Conference Acceptance Decision**

**Decision:**

Reject

**Comment:**

All three reviewers agreed that the paper was an interesting, giving a demonstration of what quantum computer could achieve. However, they all also felt that the topic was outside the main interests of the conference and better suited to other venues, e.g. a quantum computation workshop. The AC agrees with them. Thus unfortunately, the paper cannot be accepted.